# Reliability and Validity of Temporal Parameters during the Side Hop Test in Patients after Anterior Cruciate Ligament Reconstruction

**DOI:** 10.3390/jcm13123407

**Published:** 2024-06-11

**Authors:** Anna Stitelmann, Lara Allet, Stéphane Armand, Philippe Tscholl

**Affiliations:** 1Department of Orthopedic Surgery and Traumatology of the Musculoskeletal System, Geneva University Hospitals, 1205 Geneva, Switzerland; 2Geneva School of Health Sciences, HES-SO University of Applied Sciences and Arts Western Switzerland, 1202 Geneva, Switzerland; 3Wallis School of Health Sciences, HES-SO University of Applied Sciences and Arts Western Switzerland, 1950 Sion, Switzerland; 4Kinesiology Laboratory, Geneva University Hospitals and University of Geneva, 1205 Geneva, Switzerland; 5ReFORM IOC Research Centre for Prevention of Injury and Protection of Athlete Health, 4000 Liège, Belgium

**Keywords:** return to sport, ACL injury, video analysis, side hop test, temporal parameters, muscle strength, ACL–RSI

## Abstract

**Background:** The side hop test (SHT) measures the number of jumps performed over 30 s. Although this measure has demonstrated its value in clinical practice, the temporal parameters of the SHT allow for a deeper analysis of the execution strategy. The aim of this study is to assess the reliability and construct validity of contact time parameters during the SHT recorded by a video analysis system in anterior cruciate ligament reconstructed (ACLR) patients. **Methods:** We investigated the reliability (intra-rater, standard error of measurement (SEM), and minimum detectable change (MDC)), discriminant validity (operated (OP) versus non-operated (NOP) side), and convergent validity (relationship with strength and psychological readiness) of SHT contact time parameters, number of valid hops and limb symmetry index (LSI) in 38 ACLR patients. Contact time parameters are presented as mean, standard deviation (SD), and coefficient of variation (CV) of contact time. **Results:** Intra-tester reliability was good to excellent for all contact time parameters. For discriminant validity, the mean and SD contact times of the OP leg were significantly longer than those of the NOP leg, although the difference was smaller than the SEM and MDC values. The number of valid jumps and CV contact time parameters were not significantly different. Isokinetic quadriceps strength (60°/s) was strongly correlated with mean contact time for both legs. However, psychological readiness was not correlated with any of the contact time parameters. **Conclusions:** Temporal parameters of the SHT measured on video analysis are valid and reliable parameters to assess the performance strategy of the SHT. The results should be interpreted with caution regarding the SEM and MDC values. Further studies are needed to measure criterion validity, inter-rater reliability, and responsiveness.

## 1. Introduction

ACL injuries are among the most common and significant knee injuries in sports. These injuries are particularly common in young and active individuals participating in sports that involve jumping, pivoting, and changing direction, such as football and basketball [1]. Approximately 3% of amateur athletes sustain an ACL injury each year [2]. Return to sport (RTS) is the primary goal following anterior cruciate ligament reconstruction (ACLR) [3]. In amateur athletes, only 65% of the patients return to their pre-injury level of sport two years after ACLR [4]. The risk of a second ACL injury in athletes is high, up to 20–23% [5]. RTS decision-making after ACLR is challenging, as current evidence shows a conflicting relationship between RTS criteria and the potential risk of a second ACL injury [6,7,8]. For example, Webster and Hewett (2019) found no significant association between passing the return to sport (RTS) criteria and a reduced rate of subsequent ACL injury [7]. However, Capin et al. (2019) re-analyzed the data, excluding two inappropriate studies, and found that athletes who met the RTS criteria had a 75% lower risk of ACL injury than those who did not meet the criteria [6]. A criterion-based approach has also become mandatory [9,10,11]. In order to assess readiness for RTS, important functional parameters should be measured, such as functional capacity through lower limb strength, psychological readiness, hop testing, and measurement of movement quality [10,12]. Indeed, greater quadriceps strength is associated with effective RTS and a reduction in subsequent knee injuries, and psychological readiness is known to be a reason for not returning to play in 65% of cases [13,14,15,16]. Meeting qualitative movement criteria is associated with a lower rate of second ACL injury when returning to pivoting sports [17]. With regard to hop tests, the available evidence suggests a lack of consistency in their ability to predict safe RTS [18].

The side hop test (SHT) consists of jumping on one leg from side to side for 30 s between two parallel bands 40 cm apart [19]. The maximum number of hops is recorded and usually compared to the non-operated (NOP) limb and described by the limb symmetry index (LSI). Several relevant aspects are tested during the SHT, such as plyometric and side cutting movements with an endurance component, both of which are known to be highly associated with ACL injury risk due to transient poor knee control in a relatively straight, abducted, and rotated knee position [20]. As described for other hop tests, the use of the number of hops alone does not provide complete information, as it does not guarantee unaltered or symmetrical jump biomechanics and does not consider variables related to the test execution strategy, such as spatiotemporal parameters that are not obvious to the clinician’s visual perception [21,22]. Currently, there are very few studies evaluating spatiotemporal parameters during hop tests [23,24,25]. Mani et al. (2017) showed significant discriminant validity (between the involved and uninvolved sides) for both mean contact time and hop distance for a hop test consisting of four consecutive forward hops [23]. Ahmadian et al. (2020) showed no significant difference in spatiotemporal parameters during a triple hop test (consisting of three consecutive forward hops), although they demonstrated differences in hop biomechanics between healthy and injured participants [24]. To our knowledge, there are no studies on the psychometric properties of these parameters for clinical use.

The aim of this study is to investigate the reliability and validity of temporal parameters of the SHT using a video analysis system in ACLR patients. We hypothesized that the operated (OP) leg would have a longer mean contact time and greater variability in temporal parameters compared to the NOP leg and would not be associated with LSI values of the number of valid hops. Patients with a longer mean contact time or higher variability will show a moderate correlation with the strength and the ACL–RSI scale.

## 2. Methods

The method of the study was designed according to recommendations of the consensus-based standards for the selection of health status measurement instruments (COSMIN) recommendations regarding the measurement of construct validity and reliability [26]. The reporting of observational studies in epidemiology (STROBE) guidelines were used to report cross-sectional studies [27].

The ACLR in-house database was used for this retrospective study. Patient records were only analyzed if they underwent a standardized test battery between 6 and 9 months post-operatively to determine their ability to RTS [28]. Participants underwent conventional rehabilitation between the surgery and the test battery, usually consisting of two physiotherapy sessions per week. All study procedures were approved by the Research Ethics Committee of the Canton of Geneva (2020-02201).

### 2.1. Participants

Inclusion criteria were ages between 16 and 50 years, RTS testing at 6–9 months post ACLR, ability to jump, and the availability of the full data set, including videotape, isokinetic strength, and ACL–RSI assessment. This age range captures the demographic most affected by ACL injuries, spanning late adolescence to mid-adulthood, thus minimizing confounding factors. All types of graft-type surgery were accepted as they do not significantly affect performance and RTS rates after ACLR [29]. Exclusion criteria were previous knee surgery or contralateral knee surgery, inability to jump (<50% LSI of quadriceps strength), pregnancy, and >25% of invalid hops according to test guidelines [19]. The flow chart of participants is shown in Figure 1. As 50 participants were required to meet the COSMIN recommendations for reliability, 12 additional participants were selected from the database [26].

### 2.2. Return to Sport Test Protocol

Height and weight were recorded at the start of the procedure. The protocol consisted of a validated strength assessment, hop tests, and the ACL–RSI questionnaire [28]. The warm-up was started directly on the isokinetic measurement device and consisted of 5 min of concentric movement in knee extension and flexion with a difficulty of 3/10. The pre-injury level of sports activity was defined according to Hefti et al. (1993) [30]. Level I corresponds to sports with jumping, cutting, and pivoting (i.e., football, basketball), Level II corresponds to lateral movements in sports activities (i.e., volleyball, tennis, dance, martial arts), and Level III refers to sports without pivoting actions (i.e., running).

#### 2.2.1. Assessment of the 30s-SHT

Participants were asked to perform the SHT according to Gustavsson et al. (2006) [19]. After a trial run for each leg, the patients were instructed to hop as many times as possible over two lines 40 cm apart. They were instructed to start from a single-leg position and not to touch the line during the hops; their hands were free throughout the test. Patients performed the test on each leg, starting with the NOP leg, with 2 min rest in between. An invalid hop was defined as landing on or within the two lines of the tape.

The SHT was recorded with a video camera (Sony HDR-HC9E, 50 frames per second, 4.6 megapixels, Sony Corporation, Tokyo, Japan). The height of the tripod stand was adjusted to capture the knee in a standing position.

Video data were analyzed using Dartfish motion analysis software (Team Pro 7.0 version, Alpharetta, GA, USA) to calculate the number of jumps and temporal parameters. Initial contact time and toe-off events were manually recorded by applying time tags. Contact time was considered as the time between the two events (from initial contact time to toe-off event). The temporal parameters were the mean, the standard deviation (SD), and the coefficient of variation (CV) of the contact time of all jumps performed over 30 s. Contact time was based on the initial contact and toe-off events.

If adjustment jumps were performed due to instability on one side before going to the other side, the contact time was extended to reflect the full time spent on the same side. These data were exported into an Excel file (csv), which allowed us to extrapolate the mean, the SD, and the CV of the contact time. The SD gives valuable information about the dispersion and relative variability of the values, especially when the number of hops or the mean are similar between legs. The coefficient of variation was calculated from the following formula: CV = (SD/Mean) × 100(1)

The LSI of the number of valid hops and the temporal parameters were defined as the ratio between the two legs calculated as follows:LSI = (OP leg value/NOP leg value) × 100.(2)

The LSI difference was considered as: LSI difference _valid hops_ = LSI value − 100(3)

For the temporal parameters, the LSI difference was calculated as follows, as the results were expected to be higher in the OP leg:LSI difference _temporal parameters_ = (NOP leg value − OP leg value) × 100/NOP leg value(4)

The endurance components of the SHT were also assessed. The results of the number of jumps and contact time parameters were divided into two parts and compared for analysis: the first 10 s and the last 10 s of the SHT.

An example of data collection using Dartfish software (version 10.15.219.0) is provided in the Appendix A.

#### 2.2.2. Assessment of Maximal Voluntary Strength

Participants were asked to perform one trial for each leg at submaximal intensity prior to the test using the Con-Trex MJ isokinetic dynamometer (Con-Trex MJ, CMV AG, Dübendorf, Switzerland). During the test, participants were restrained with straps around the trunk, pelvis, and thigh to minimize compensatory movements. The rotation axis of the torque meter was aligned with the femoral condyle of the tested limb. The knee range of motion was 90° (from 100° to 10° of knee flexion and 0° corresponding to the fully extended knee). The warm-up and test procedure are detailed in Appendix B. The result is the peak torque to weight ratio (N·m·kg^−1^) developed during concentric movement at a speed of 60°/second of the quadriceps and hamstrings and during eccentric movement of the hamstrings at a speed of 90°/second. The NOP limb was tested first, and encouragement was given during the measurements.

#### 2.2.3. Assessment of Psychological Readiness (ACL–RSI)

The ACL–RSI scale was developed to assess an athlete’s psychological readiness to return to sport after ACLR surgery [31]. The questionnaire consists of twelve questions to which patients score their responses from 0 to 100 with 10-point increments. The total score is calculated as the average of all responses. Higher scores indicate greater psychological readiness. The French version of the ACL–RSI has been previously validated for patients after ACLR [32].

### 2.3. Statistical Analysis

All results are presented as median ± interquartile range (IQR).

#### 2.3.1. Intra-Rater Reliability, SEM, and MDC

Reliability measurements were performed twice by the first author (with at least 4 weeks of interval). The tester was blinded for the information of the operated leg and repeated the measurements with a 4-week interval. Fifty participants were included for reliability, as this is the sample size recommended by the COSMIN guidelines [26]. Intra-tester reliability of the SHT contact time parameters was assessed using intraclass correlation coefficients (ICC (3,1)) and 95% confidence intervals (CI). A two-way mixed effect model was used, with the type being considered as a single measure and definition as absolute agreement. ICCs above 0.9 were considered excellent, between 0.75 and 0.9 as good, between 0.5 and 0.75 as moderate, and <0.5 as poor [33].

The standard error of measurement (SEM) is considered to be the SD of the variance of the test within individuals. It is calculated as SD * √(1 − ICC). The minimal detectable change (MDC) is the amount of observable change required to exceed the expected measurement error of the within-subject variability. It is calculated as 1.96 * SEM.

#### 2.3.2. Construct Validity

An a priori power calculation of ten patients was performed to determine the sample size. A total of 31 participants are required to achieve 80% power and 5% significance level to detect a mean of the differences between pairs of 112.9, assuming a standard deviation of the differences of 238.6 [34]. As the normality of the samples was rejected for the parameters studied using the Shapiro–Wilk normality test, non-parametric statistical tests were used. The Wilcoxon signed-rank test was used to assess the discriminant validity (OP vs. NOP side) of the temporal parameters and the number of jumps. Convergent validity was further assessed by correlating temporal parameters and number of jumps with limb strength and ACL–RSI score using Spearman’s correlation coefficients. Correlation coefficients were considered as very weak (0 ≤ |r| ≤ 0.19), weak (0.2 ≤ |r| ≤ 0.39), moderate (0.4 ≤ |r| ≤ 0.59), strong (0.6 ≤ |r| ≤ 0.79), and very strong (0.8 ≤ |r| ≤ 1) [35]. Statistical significance was set at *p* < 0.05. Statistical analyses were performed using R studio software (Desktop, 2022.02.2-485, version 1.4.1717).

## 3. Results

Patient characteristics are described in Table 1. The median age of 26.4 years shows that the central age of the participants is in the mid-20s. The IQR of 21.6 to 35.1 years indicates that the age of the middle 50% of participants is spread over a range of 13.5 years. This range shows moderate variability, suggesting that although there is some diversity in age, most participants are relatively young adults. Thirty-eight ACLR patients were evaluated at 7.5 months [6.1–9.0] post-operatively. Twenty-one percent of the participants were women. Ninety-two percent of participants received a graft from their ipsilateral extensor apparatus. The different sports played by the participants prior to injury are described in Appendix C.

### 3.1. Intra-Rater Reliability, SEM, and MDC of Temporal Parameters

The results of intra-tester reliability, SEM, and MDC are shown in Table 2. All contact time parameters showed excellent intra-tester reliability (0.98–0.99) except for the SD contact time of the OP leg, which showed good intra-tester reliability (0.87). The SEM and MDC were higher in the OP leg than in the NOP leg.

### 3.2. Discriminant Validity of Temporal Results

#### 3.2.1. Side-to-Side Differences and Corresponding LSI

The side differences and the corresponding LSI difference values are presented in Table 3. No significant leg differences were found for the number of valid hops. However, the mean contact time of the OP leg was significantly longer compared to the NOP leg (*p* = 0.011) for both medial and lateral contact time (*p* = 0.021 and *p* = 0.015, respectively). In terms of contact time variability, the OP leg had a significantly greater overall SD contact time than the NOP leg (*p* = 0.042). High interpatient variability, as indicated by large SDs, was observed in both legs for the number of valid jumps and all temporal parameters.

The LSI differences of the valid hops (−7.5%) were compared with the LSI differences of the mean, SD, and CV contact times (−10.8, −17.8, and −1.3%, respectively). The LSI differences of the mean and SD contact times were statistically different from the LSI differences of the valid hops (*p* = 0.008 and *p* = 0.039, respectively).

The discriminant validity between medial and lateral hops for all parameters and both legs showed no statistical difference.

#### 3.2.2. Endurance Components

The endurance components of the number of valid jumps and the temporal parameters are presented in Table 4. For both legs, the number of valid jumps was significantly lower, and the mean contact time was significantly higher during the last ten seconds of the SHT than during the first ten seconds (*p* < 0.05). In terms of variability (SD and CV contact time), SD and CV were greater during the last ten seconds of the SHT than during the first ten seconds, but no significant difference was found between the legs. The analysis revealed no difference between the OP and NOP legs for the number of valid hops and all contact time parameters.

### 3.3. Convergent Validity between Strength and Temporal Parameters

Patients had a lower LSI value for quadriceps strength (77.4%) than for hamstring strength (92.9% and 89.0%, respectively). The participants’ limb strength scores are presented in Table 5. Convergent validity results for the number of valid hops and contact time parameters related to the strength are presented in Table 6. Correlations with all strength variables were strong for the number of valid hops, mean and SD contact time, and moderate for CV contact time. The highest correlation observed for mean and SD contact time was with quadriceps strength. The OP leg showed a higher correlation with strength than the NOP leg.

### 3.4. Convergent Validity between ACL–RSI and Temporal Parameters

The mean ACL–RSI score of the participants was 69 (±16). The discriminant validity between the ACL–RSI scale and the temporal parameters is presented in Table 7. For the ACL–RSI score, no correlation was found between any of the variables. 

### 3.5. Potential Co-Founding Factors from Participant’s Characteristics

The height and BMI values were compared with the number of valid hops and all contact time parameters. The results are presented in Table 8. Height and BMI were not confounding factors, as no strong correlation was found between all parameters. The correlation was only significant in both legs when CV contact time was compared with the height of the participants.

## 4. Discussion

This study is the first to analyze temporal parameters of the SHT, such as variability, and compare them with other important clinical parameters, such as leg strength, using an affordable and clinician-friendly 2D video analysis system.

The aim of this study was to investigate the reliability and validity of temporal parameters of the SHT in ACLR patients. Measures of temporal parameters of the 30 s SHT showed good to excellent intra-tester reliability. As discriminate factors, mean and SD contact times were significantly longer in the OP leg compared to the NOP leg. These results should be interpreted with caution, as the mean difference did not exceed the MDC and SEM values. This means that the observed change may be partially explained by measurement error or random variation. However, the number of valid hops and CV contact time had no discriminative value. Quadriceps and hamstring strength showed a strong to moderate negative correlation with mean and SD contact time for both legs, whereas psychological readiness was not correlated with any contact time parameter.

The results of this study revealed differences in performance between the OP and NOP in terms of strategies for performing the SHT. Temporal parameters have better psychometric properties than the number of valid jumps. The number of jumps had no discriminating value, whereas the temporal parameters seem to be more discriminating and sensitive to changes.

The average number of valid jumps (35–56 hops) and LSI score (93–99%) reported in the literature are similar to this study [36,37,38,39,40,41,42]. However, performance is not in line with healthy normative values (55–57 jumps with an average LSI value of 100%) [42].

Participants in this study had difficulty maintaining a consistent rhythm during the SHT on their OP, as the overall SD and corresponding LSI difference were able to discriminate limb differences. These results show that the stabilization time on the OP leg can vary from jump to jump; although some jumps were fast, others required a longer stabilization time. A longer stabilization time can be explained by an adjustment jump or a loss of balance. This means that participants had more difficulty maintaining a consistent rhythm on their OP leg during the SHT. However, the patient’s strategy can significantly influence the results. If the pace is slowed down or more invalid hops are made to reduce the difficulty, the results in this study may be less variable than expected.

When comparing the endurance components of the SHT, a similar decrease in temporal parameters was recorded for both legs, with no leg difference when comparing the performance of the first and last ten seconds. It is possible that the duration of the test is not long enough to cause fatigue and discriminate between the two legs in terms of endurance components. 

In this study, the inter-individual differences between valid hops and temporal parameters were very high. Several patients performed some adjustment jumps (several small hops on one side), which significantly increased the value of the contact time parameters. This indicates a poor stabilization strategy and, therefore, poor motor control or fatigue.

In terms of convergent validity, low scores on the number of valid hops and higher mean and SD contact times were associated with low quadriceps or hamstring strength. The correlation was slightly stronger for the quadriceps, probably because most of the patients received a quadricipital or patellar graft on the OP leg, and therefore, the strength for this muscle was lowest. These findings are consistent with those of Thomas et al. (2015), who found that maximum force production during isometric mid-thigh pull correlated with change of direction (COD) ability in athletes [43].

However, the temporal parameters of the SHT were not correlated with subjective psychological readiness. As the questionnaire is not specific to the SHT task but to return to a sport defined by the patient, it would be interesting to be more specific and directly measure anxiety during the SHT and the level of anxiety related to specific tasks such as cutting maneuvers.

In terms of confounders, it is interesting to note that the participant’s height was weakly correlated with CV contact time in both legs, suggesting that reduced height increases variability in contact time due to increased difficulty in performing the test. Higher variability may be the result of greater difficulty in performing the test. It would, therefore, be interesting to propose a SHT with a width between the bands that is proportional to the size of the participants’ lower limbs.

If we compare these results with the study by Urhausen et al. (2022), they found no significant differences in the number of jumps or temporal parameters [25]. However, the characteristics of their participants are different from those in this study. The SHT in our study was performed, on average, 1.4 months earlier, and our participants had a majority of quadriceps tendon grafts (instead of semi-tendinous grafts in their study). In addition, their reported rate of invalid hops (0.2%) was lower than in this study (8–9%). Probably due to the early testing, our participants perceived more difficulty in performing the test and may have a different execution strategy. On the one hand, an invalid hop may lead to a shorter contact time because the effort was less difficult than required. On the other hand, it is also possible that the effort was too difficult, and therefore, a longer stabilization time followed due to a lack of balance.

### Strength and Limitations

The first major limitation is the large standard deviation of all contact time results, which reflects the heterogeneity of SHT performance in the population of this study, although patients with too many invalid jumps were excluded from the analysis. 

The second main limitation of this study was that the criterion validity of the video measurement was not measured. It is possible that the accuracy of the measurement device does not allow enough difference in the temporal parameters between the legs to determine adequate variability. The mean would probably not change much, and the SD would remain high, but the coefficient of variation is very sensitive to variations in numbers. Temporal parameters are probably more important than the number of valid jumps performed. However, a standard 50Hz camera may be inadequate, as the potential error is 8% due to the large time frame. The 8% is explained by the difference in measurement when the evaluator hesitates between 2 frames to determine the 1st ground contact. Therefore, better cameras with 200 frames/second could improve the measurement accuracy.

In addition, the average time taken to obtain time parameter data from an individual in this study was around 10 min. The use of other tools, such as inertial measurement units, force plates, or even artificial intelligence, would significantly reduce the measurement time, which would be of great benefit to clinicians. An in-depth analysis of the measurement properties of this tool should be carried out.

Another potential bias in this study was the use of a paired statistical test. As the results compared data of both legs of each individual, the data were considered as paired. However, the assumption of symmetry between the legs is questionable. Differences such as the presence of a dominant leg or medical history can influence asymmetry. 

Finally, as the participants had no pre-operative values, this study assessed the limb asymmetry of participants having undergone unilateral ACLR regarding the number of valid jumps and temporal parameters. However, deficit after ACLR is also common in the NOP leg, and the limb symmetry index may overestimate the hop performance [44]. This makes it difficult to assess the responsiveness of the test. Further studies are needed to assess these parameters in healthy control subjects or compare the results with pre-operative values. 

## 5. Conclusions

Contact time parameters during the SHT can be reliably measured using a video analysis system. Mean and SD contact time parameters were able to discriminate between the OP and NOP leg in ACLR patients, whereas the number of side hop jumps had no discriminating value. However, the interpretation of temporal parameters requires some caution due to the MDC and SEM values. This demonstrates that temporal parameters can give an indication of the patient’s execution strategy during the SHT. Finally, psychological readiness did not influence contact time parameters, whereas low quadriceps and hamstring strength were associated with longer mean contact time and higher variability. Further studies are needed to measure its criterion validity, inter-rater reliability, and responsiveness. 

## Figures and Tables

**Figure 1 jcm-13-03407-f001:**
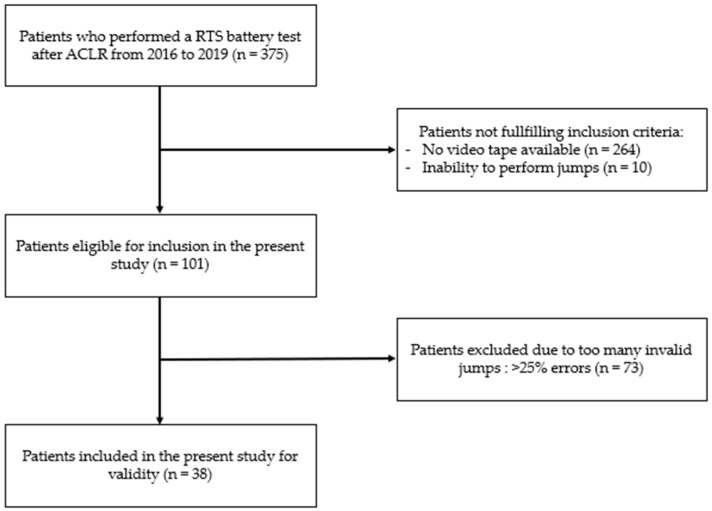
STROBE flow chart of participants. The abbreviations used are as follows: RTS: return to sport; ACLR: anterior cruciate ligament reconstruction; n: number; %: percentage.

**Table 1 jcm-13-03407-t001:** Characteristics of participants (*n* = 38).

	Characteristics of Participants
Gender (W/M) (n)	8/30
Age (years)	26.4 [21.6–35.1]
BMI (kg/m^2^)	23.8 [21.4–26.2]
Injured leg (R/L)	26/12
Sport activity level pre-injury ^1^	I (*n* = 28)/II (*n* = 8)/III (*n* = 2)
Time post-surgery (months)	7.5 [6.1–9.0]
Graft type (n)	QT = 32; STG = 3; BPTP = 3
Associated lesions (n)	MM = 7; LM = 7; both = 6; none = 18

Variables are presented as median ± IQR. n: Number of participants; W: Woman; M: Man; BMI: Body Mass Index; R: Right; L: Left; QT: Quadriceps tendon; STG: Semitendinosus/gracilis graft tendon; BPTP: Bone-patellar Tendon-bone; MM: Medial meniscus; LM: Materal meniscus. ^1^ According to Hefti et al. 1993 [25].

**Table 2 jcm-13-03407-t002:** Intra-rater reliability, SEM, and MDC of temporal parameters (*n* = 50).

Variables	Test	Retest	ICC (95%CI)	*p*-Value	SEM	MDC
Mean contact time (ms)						
OP (ms)	452 [343–703]	461 [331–685]	0.98 (0.96–0.99)	<0.001	73	202
NOP (ms)	385 [311–514]	388 [317–520]	0.99 (0.99–0.99)	<0.001	35	98
SD contact time (ms)						
OP (ms)	114 [49–273]	113 [50–372]	0.87 (0.79–0.92)	<0.001	98	271
NOP (ms)	84 [57–202]	81 [53–204]	0.99 (0.99–1.00)	<0.001	21	59
CV contact time (%)						
OP (%)	25.9 [13.7–37.7]	26.0 [15.7–37.3]	0.99 (0.99–1.00)	<0.001	1.8	4.9
NOP (%)	22.5 [17.1–31.1]	21.0 [16.3–35.1]	0.99 (0.98–0.99)	<0.001	1.8	4.6

Variables are presented as median ± IQR. OP: Operated leg; NOP: Non-operated leg; SD: Standard deviation; CV: Coefficient of variation; ms: Milliseconds; %: Percentage.

**Table 3 jcm-13-03407-t003:** Discriminant validity of temporal parameters and corresponding LSI differences (*n* = 38).

Variables	OP	NOP	*p*-Value	LSI Differences (%)
Number of hops				
Valid hops	41 [32–54]	43 [33–54]	0.113	−10.8 [−25.2–1.5]
% invalid hops	8.6 [3.1–13.7]	7.3 [4.0–13.1]	0.374	-------------------------
Mean contact time (ms)				
Overall	438 [328–733]	430 [323–608]	**0.011**	−10.8 [−25.2–1.5]
Medial	442 [324–774]	431 [316–591]	**0.021**	−9.4 [−26.1–3.5]
Lateral	452 [327–737]	403 [325–572]	**0.015**	−10.0 [−18.6–2.7]
Standard deviation contact time (ms)				
Overall	133 [50–404]	132 [63–246]	**0.042**	−17.8 [−64.3–21.3]
Medial	114 [50–326]	106 [54–289]	0.155	−10.8 [−75.6–29.2]
Lateral	109 [49–276]	123 [55–196]	0.229	−2.8 [−53.6–22.3]
Coefficient of variation contact time (%)				
Overall	28.4 [15.0–43.8]	28.1 [18.1–39.7]	0.523	−1.3 [−43.4–24.2]
Medial	25.8 [14.8–40.6]	24.1 [16.6–35.9]	0.760	6.5 [−35.6–26.8]
Lateral	22.3 [13.7–42.8]	24.5 [15.7–40.1]	0.896	4.7 [−29.2–28.8]

Variables are presented as median ± IQR. n: Number of participants; OP: Operated leg; NOP: Non-operated leg; LSI: Limb symmetry index; ms: Milliseconds.

**Table 4 jcm-13-03407-t004:** Discriminant validity of temporal parameters according to the first 10 s (F) and last (L) 10 s time frame of the SHT (*n* = 38).

Variables	OP		NOP		*p*-Value		
	First	Last	First	Last	OP	NOP	Leg Effect
Valid hops	15 [11–18]	12 [8–16]	17 [15–20]	14 [9–17]	**<0.001**	**<0.001**	0.940
Mean (ms)	427 [329–654]	504 [326–792]	425 [315–498]	466 [317–742]	**0.013**	**0.002**	0.862
SD (ms)	119 [60–220]	136 [39–410]	76 [50–159]	113 [51–262]	0.500	**0.035**	0.892
CV (%)	21.7 [16.1–34.4]	24.2 [12.3–40.2]	16.8 [12.7–31.4]	21.4 [14.8–42.0]	0.942	0.264	0.629

Variables are presented as median ± IQR. s: Seconds; OP: Operated leg; NOP: Non-operated leg; SD: Standard of deviation; CV: Coefficient of variation; Diff: Difference; First: First 10 s of side hop test results; Last: Last 10 s of side hop test results.

**Table 5 jcm-13-03407-t005:** Strength peak torque to weight ratio values of participants (*n* = 38).

Strength (N·m·kg^−1^) ^1^	OP	NOP	LSI (%)
Concentric Q—60°/s	1.81 [1.52–2.21]	2.47 [2.12–2.79]	77.4 (± 18.2)
Concentric HS—60°/s	1.30 [1.08–1.57]	1.50 [1.21–1.65]	92.9 (± 13.4)
Eccentric HS—90°/s	1.58 [1.29–1.85]	1.73 [1.53–1.98]	89.0 (± 21.4)

Variables are presented as median ± IQR. OP: Operated; NOP: Non-operated; Q: Quadriceps; HS: Hamstrings; LSI: Limb symmetry index. ^1^ Peak torque variables measured with isokinetic concentric or eccentric testing with 5 repetitions at an angular velocity of 60 or 90° per second.

**Table 6 jcm-13-03407-t006:** Correlation between strength components and temporal parameters (*n* = 38).

Variables	OP		NOP	
	R	*p*-Value	R	*p*-Value
Strength (N·m·kg^−1^) between valid hops				
Concentric Q—60°/s	0.74	<0.001	0.61	<0.001
Concentric HS—60°/s	0.68	<0.001	0.56	<0.001
Eccentric HS—90°/s	0.68	<0.001	0.53	<0.001
Strength (N·m·kg^−1^) between overall mean contact time (ms)				
Concentric Q—60°/s	−0.73	<0.001	−0.60	<0.001
Concentric HS—60°/s	−0.67	<0.001	−0.53	<0.001
Eccentric HS—90°/s	−0.62	<0.001	−0.51	0.001
Strength (N·m·kg^−1^) between overall SD contact time (ms)				
Concentric Q—60°/s	−0.72	<0.001	−0.60	<0.001
Concentric HS—60°/s	−0.69	<0.001	−0.51	<0.001
Eccentric HS—90°/s	−0.64	<0.001	−0.45	0.004
Strength (N·m·kg^−1^) between overall CV contact time (%)				
Concentric Q—60°/s	−0.56	<0.001	−0.54	<0.001
Concentric HS—60°/s	−0.60	<0.001	−0.43	0.008
Eccentric HS—90°/s	−0.59	<0.001	−0.35	0.034

OP: Operated leg; NOP: Non-operated leg; R: Spearman’s correlation coefficient; Q: Quadriceps; HS: Hamstring

**Table 7 jcm-13-03407-t007:** Comparison between ACL–RSI scale and valid hops/temporal parameters of the operated leg (*n* = 38).

Variables	R	*p*-Value
Valid hops	0.04	0.814
Mean contact time (ms)	−0.05	0.767
SD contact time (ms)	−0.15	0.364
CV contact time (%)	−0.19	0.266

R: Spearman’s correlation coefficient; ACL–RSI: anterior cruciate ligament–return to sport injury scale; SD: standard deviation; CV: coefficient of variation; ms: milliseconds; %: percentage.

**Table 8 jcm-13-03407-t008:** Correlation between valid hops/temporal parameters and participant’s characteristics (*n* = 38).

Variables	OP		NOP	
	R	*p*-Value	R	*p*-Value
Height between valid hops and temporal parameters				
Valid hops	0.33	**0.046**	0.26	0.117
Mean CT (ms)	−0.24	0.146	−0.26	0.116
SD CT (ms)	−0.31	0.062	−0.29	0.080
CV CT (%)	−0.33	**0.045**	−0.34	**0.037**
BMI between valid hops and temporal parameters				
Valid hops	−0.20	0.227	−0.28	0.087
Mean CT (ms)	0.29	0.073	0.32	**0.046**
SD CT (ms)	0.28	0.090	0.33	**0.045**
CV CT (%)	0.16	0.339	0.23	0.174

OP: Operated leg; NOP: Non-operated leg; R: Spearman’s correlation coefficient; BMI: Body mass index; CT: Contact time; ms: Milliseconds; %: Percentage.

## Data Availability

The original data presented in the study are openly available.

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
