# Peer review of "Reliability and Validity of Temporal Parameters during the Side Hop Test in Patients after Anterior Cruciate Ligament Reconstruction"

_jcm, 2024, doi:10.3390/jcm13123407_

Round 1

Reviewer 1 Report

Comments and Suggestions for Authors

Dear Author's

Thank you for the opportunity read the results Your article. Unfortunately, I have many doubts, I provides them below:

- “Return to sport (RTS) is the main goal after anterior cruciate ligament reconstruction 46 (ACLR)”. – where does such a generalizing statement come from? do other people/patients suffer from this type of injuries?, please explain... it seems that the authors of the paper believe that ACL injuries concern athletes and no one else... is this how it should be understood? - what group of patients does the work concern? (playing sports)??? – it's not clear - please expand the introduction with general data on ACL injuries - “undergone a standardized testing battery” - please explain what tests you mean... - was the type of surgery performed taken into account when selecting the research group / is it very important? - was possible rehabilitation after the procedure taken into account when selecting the research group? - assessment 6-9 months after surgery (what happened to these people/patients before the assessment) - no data available? (I suspect that this is a heterogeneous group of patients and the results obtained may be “very inaccurate”… - statistical analysis - please describe the statistical analysis step by step, what tests were used - Wilcoxon test to assess OP vc NOP side - this is a gross error...? - please complete the effect size and sample size (not included in the work) - references – some items are over 10-15 years old (1996, 2006, 2008) - age 16 - 50, was the age selection dictated / was there a large age discrepancy?, please explain... - 38 patients – a non-representative group – shouldn't it be a pilot study? - what was the average test execution time, who conducted them? - discussion - lack of any discussion - reference to practically one other study - to be improved - limitations - the fact that this is the first study is not a limitation... - limatations to - lack of a control group, sample size calculations, etc. ... and not a statement that "in-depth research" is needed ...

  best regards for all Author's

Reviewer 2 Report

Comments and Suggestions for Authors

1. RTS decision making after ACLR is challenging as current evidence shows a conflicting relationship between RTS criteria and possible risk for 2nd ACL injury - can the authors elaborate on these findings? They seem a little bit vague.

2. important functional parameters should be measured - define each functional parameter and its importance in evaluating RTS readiness

3. very few studies assessing spatiotemporal parameters during hop tests - can the authors cite some studies and elaborate on these studies?

5. Authors must elaborate on the dataset they used. Please discuss on the database, including the type of data it contains and how patients were selected for analysis.

6. What was the warm-up in the isokinetic device?

7. A concise summary of the SHT protocol should be provided.

8. The use of each statistical test should be explained, together with what data was used on.

Comments on the Quality of English Language

4. There are English errors, out of which I shortly identified the following:

·      Line 85: "between 6 to 9 months" should be "between 6 and 9 months"

·      Line 89: "age between 16 and 50 years" should be "ages between 16 and 50 years"

·      Row 103: "measured" must be "recorded".

·      Row 111: " separated by 40cm" space is missing "40 cm".

·      Line 118: "4.6 Megapixel" should be "4.6 megapixels"

Authors should consider that I have only highlighted few of the errors seen. There are many others, and the manuscript requires a good English proofread before being published. I understand that English might not be the native language, but a scientific paper should be pristine from this point of view.

Round 2

Reviewer 1 Report

Comments and Suggestions for Authors

Dear Author's

Thank you for responding to my comments. I believe that the manuscript
is more valuable. I accapt it in current form.

best regards for all Author's.

Reviewer 2 Report

Comments and Suggestions for Authors

I accept the authors changes and comments. The manuscript has been greatly improved.